# Pretransplant BMI Significantly Affects Perioperative Course and Graft Survival after Kidney Transplantation: A Retrospective Analysis

**DOI:** 10.3390/jcm11154393

**Published:** 2022-07-28

**Authors:** Małgorzata Dobrzycka, Beata Bzoma, Ksawery Bieniaszewski, Alicja Dębska-Ślizień, Jarek Kobiela

**Affiliations:** 1Department of General, Endocrine and Transplant Surgery, Medical University of Gdansk, 80-210 Gdansk, Poland; ksawery.bieniaszewski@gumed.edu.pl (K.B.); kobiela@gumed.edu.pl (J.K.); 2Department of Nephrology, Transplantology and Internal Medicine, Medical University of Gdansk, 80-210 Gdansk, Poland; beata.bzoma@gumed.edu.pl (B.B.); adeb@gumed.edu.pl (A.D.-Ś.)

**Keywords:** kidney transplantation, obesity, metabolic surgery

## Abstract

Background. The number of kidney transplant recipients (KTRs) with overweight and obesity is increasing. It was shown that obesity is related to inferior patient and graft survival. We aimed to analyze intraoperative parameters and postoperative short and long-term course of kidney transplantation (KT) in body mass index (BMI)-stratified cohorts of KTRs. Methods. A retrospective analysis of a prospectively built database of 433 KTRs from 2014 to 2017 from a single transplant center was performed. The objective of the study was to analyze the association between BMI at the time of transplantation with intraoperative parameters, adverse events in early postoperative course, and the overall mortality and graft loss in BMI-stratified cohorts: normal (18.5 and 24.9 kg/m^2^), overweight (25–29.9 kg/m^2^) and obese (≥30 kg/m^2^). Results. Obesity was related to longer total procedure time (*p* = 0.0025) and longer warm ischemia time (*p* = 0.0003). The postoperative course in obese patients was complicated by higher incidence of DGF (delayed graft function), early surgical complications (defined as surgical complications <30 days from KT), reoperation rate, vascular complications, incidence of lymphocele and wound dehiscence. There was no difference between the normal weight and overweight KTRs. The one-month kidney function (*p* = 0.0001) and allograft survival (*p* = 0.029) were significantly inferior in obese patients with no difference between normal weight and overweight patients. One-year death-censored graft survival was better in patients with BMI < 30 (88.6 vs. 94.8% *p* = 0.05). BMI was a significant predictor of graft loss in univariate (*p* = 0.04) but not in multivariate analysis (*p* = 0.09). Conclusion. Pretransplant obesity significantly affects the intraoperative and postoperative course of kidney transplantation and graft function and survival. The course of transplantation of overweight is comparable to normal BMI KTRs, and presumably pretransplant weight reduction to the BMI < 30 kg/m^2^ may improve the short-term postoperative course of transplantation as well as may improve graft survival. Thus, pretransplant weight reduction in obese KTRs may significantly improve the results of kidney transplantation. Metabolic surgery may play a role in improving results of KT.

## 1. Introduction

The number of end-stage kidney disease (ESKD) patients with obesity worldwide is growing [1]. Obesity has not only been shown to be an independent risk factor for developing ESKD but also is related to the development of comorbidities that may impact the results of ESKD treatment [2]. Kidney transplantation (KT) is considered the most cost-effective therapy for ESKD [3]. Due to the obesity epidemic, the number of obese transplant candidates is rapidly increasing [1]. Recent studies have confirmed the relationship between obesity and many important comorbidities in kidney transplant recipients (KTRs) [4,5]. It has been proven that obesity is associated with significantly shorter allograft survival and higher overall mortality [6,7]. The increasing number of obese KTRs is also related to a higher incidence of late complications [1,5]. Interestingly, the association between recipient body mass index (BMI) and the intraoperative and early postoperative course has not been extensively studied.

The primary objective of the current study was to determine the association between BMI at the time of transplantation with intraoperative parameters. We also aimed to analyze the adverse events in early postoperative course, overall mortality and graft loss in BMI-stratified cohorts.

## 2. Materials and Methods

The study was a retrospective analysis of a prospectively built database of 433 patients with ESKD who received deceased donor KT in one large kidney transplantation center in Gdansk between 1 January 2014 and 31 December 2017. Clinical data were obtained from a prospectively built database of ESKD patients. All patient information was anonymized. All adult patients who underwent KT in that period of time were included. Patients were qualified for KT according to national practice guidelines and listed on a transplantation list [8]. All transplantations were performed by a group of six transplant surgeons. The vascular anastomosis of the renal artery to external iliac artery, and the renal vein to the external iliac vein were performed with continuous vascular suture. They were followed by ureterovesical anastomosis with JJ stenting. All patients received induction therapy with calcineurin inhibitor and mycophenolate mofetil (MMF). Anti-thymocyte globulin was administered when indicated. In the postoperative course, patients were treated with triple immunosuppressive protocols, including calcineurin inhibitor (tacrolimus or cyclosporine based on individual recommendation), mycophenolate mofetil (MMF) and glucocorticoids. Antibiotic prophylaxis was administered in all KTRs. CMV prophylaxis was recommended for all donor positive/recipient negative patients.

The database included both donor and recipient data. Data analysis included patient demographics (gender, age, BMI). Body weight (kg) and body height (m) were measured to calculate the BMI as weight in kg/m^2^ height. Obesity was defined by BMI of ≥30 kg/m^2^. For statistical analysis the study group was stratified by BMI into three study groups: normal BMI (between 18.5 and 24.9 kg/m^2^), overweight (BMI 25–29.9 kg/m^2^) and obese (BMI 30 kg/m^2^ and higher) according to WHO guidelines. Total procedure time, cold ischemia time (CIT), warm ischemia time (WIT) and hospitalization time were analyzed. Intraoperative, 30 day postoperative and delayed follow-up data were reviewed. Study follow-up was conducted up to 7 years. The primary cause of ESKD and most common comorbid conditions were noted. The comorbidities were identified at the time of KT (baseline) in patient clinical examination. The Charlson Comorbidity Index (CCI) was used for baseline characteristics as a predictor not only of the patient’s clinical situation, but also to demarcate differences among three cohorts of our patients sharing the same medical diagnosis (ESKD). We chose the Charlson Comorbidity index because it was designed to predict mortality and also may be used to predict future outcome or stratify patients into different prognostic groups better than the analysis of single comorbid conditions. Delayed graft function (DGF) was defined as the need for hemodialysis (HD) during the first week after transplantation. An acute rejection episode (ARE) is characterized by an acute post-transplant decline in kidney function as a consequence of an immune response of the host to the graft. Surgical complications included wound infection, hematoma and lymphocele. We separately recorded vascular complications (vascular anastomosis leak and embolism) and urological complications (ureteral injury and anastomosis leak). All analyses were performed in three above mentioned study groups stratified by BMI.

The study was approved by Institutional Ethics Board and was performed in accordance with the ethical standards (NKBBN/340/2016).

### Statistical Methods

Descriptive statistics were used to report continuous data as mean and standard deviation. Bivariate comparisons were made using the Student *t* test or the Mann-Whitney U test. Categorical data were expressed as values and percentages. Categorical data were compared with either χ2 test or Fisher’s exact test. Analysis of variance (ANOVA) or, when the assumptions for ANOVA were not fulfilled, the Kruskal-Wallis test was used to investigate the associations between BMI and postoperative adverse events in three groups, with post hoc analysis multiple comparison of mean ranks and Bonferroni correction.

Graft and patient survival curves were generated using the Kaplan-Meier estimator. Analyses of survival were performed in two groups: non-obese (BMI < 30 kg/m^2^) and obese (BMI ≥ 30 kg/m^2^). Statistically significant variables in the univariate analysis were introduced in a multivariate model with multiple logistic regression. Associations are given as odds ratios with a 95% confidence interval. The significance limit was set at 0.05. All statistical analysis was performed using Statistica 13.3 Software (TIBCO Software Inc., Palo Alto, CA, USA).

## 3. Results

A total of 433 heterotopic cadaveric KTs were performed in 272 (62.8%) men and 161 (37.2%) women during the studied period. All consecutive KTRss were enrolled into study. The mean BMI of the study group was 25.33 ± 4.2 kg/m^2^. Patients with obesity (BMI ≥ 30 kg/m^2^) constituted 16.6% of the study group. The obese cohort consisted of 72 patients (42 males, 30 females), with an age range of 20–74 (mean 53.4 ± 13) years. Mean age of the study group was 49.3 ± 13.8 years, with a statistical difference between obese versus normal weight and obese versus overweight (*p* < 0.05). There were more men in the overweight group (*p* = 0.002). The most common comorbidity was hypertension, observed in 322 (74.3%) of KTRs. Only the incidence of diabetes mellitus was different between the analyzed BMI stratified groups (*p* = 0.001). The mean CCI was significantly higher in the obese group, in relation to both overweight and normal weight subjects (*p* = 0.029) (Table 1).

There was no difference in dialysis modality before KT between groups (Table 1). The most common form of dialysis in all groups was hemodialysis, accounting for 76% of all cases.

The underlying renal diseases in the studied group included chronic glomerulonephritis (35.6%), chronic interstitial nephritis (11.6%), diabetic nephropathy (12.3%), polycystic kidney disease (13.2%), hypertensive nephropathy (8.9%), and “not known” or “other” in about 18.4% of patients. Other causes of ESKD included hemolytic uremic syndrome, amyloidosis, ESKD after chemotherapy and nephrectomies because of cancer. Mean donor age was 48.6 years (*p* = 0.05), and mean BMI was 25.9 kg/m^2^ (*p* = 0.08). There was a significant difference in both donor-recipient weight and BMI ratio, with the difference between analyzed cohorts (for BMI ratio *p* = 0.00, normal vs. overweight *p* = 0.00, normal vs. obese *p* = 0.00, overweight vs. obese *p* = 0.0001, for weight ratio *p* = 0.00, normal vs. overweight *p* = 0.00, normal vs. obese *p* = 0.00, overweight vs. obese *p* = 0.013). The right kidney was transplanted in 220 patients (50.9%) and the left kidney in 213 patients (49.1%). Single graft renal artery was present in 345 patients (79.7%). The standard immunosuppressive protocol included calcineurin inhibitor: tacrolimus in 228 patients (52.7%), everolimus in 14 patients (3.2%), cyclosporine in 204 patients (47.1%); MMF in 394 patients (91%) and glucocorticoids in all patients. Induction therapy with anti-thymocyte globulin was administered in 82 patients (18.9%).

The detailed characteristics of all studied groups of patients after kidney transplantation and the comparison between BMI cohorts are presented in Table 1.

Characteristics and comparison of intraoperative variables in the studied groups.

The average total time of procedure of KT was 181.98 ± 37.6 min, with a median of 180 min. We found significant difference in total procedure time between the three cohorts: normal BMI, overweight and obese (*p* = 0.0025). Mean and median values of total procedure time are presented in Table 1. The comparison of duration of the procedure between BMI cohorts is presented in Figure 1. The longest mean time of the procedure was observed in patients with obesity. A statistically significant difference was observed between the obese and normal weight group, no difference was observed between the normal weight group and overweight group.

Cold ischemia time (CIT) did not differ between the groups, whereas warm ischemia time (WIT) was significantly longer in the patients with obesity (Figure 2). The mean warm ischemia time was 27.827 ± 9.38 min, with a median of 26 ± 4.5 min. There was a difference between obese and overweight KTRs and obese and normal weight KTRs (*p* = 0.0003), whereas no difference was found between normal weight and overweight KTRs (Table 1 and Figure 2).

In a gender subset comparison, obese and overweight male KTRs had significantly longer total procedure time and WIT (*p* = 0.039 and *p* = 0.007, respectively). This relationship was not present in female recipients (*p* = 0.29 and *p* = 0.42, respectively). No intraoperative deaths were reported. There was no difference in total hospitalization time between the analysed groups (*p* = 0.15).

The incidence of postoperative adverse events and early kidney function in BMI stratified kidney transplant recipients included forty-two cases (9.7%) of acute rejection episodes (ARE) and 144 cases (33.26%) of delayed graft function (DGF). The incidence of ARE was not different among obese, overweight, and normal weight KTRs (11.4%, 8.5%, 10.04% respectively, *p* = 0.36).

The groups differed significantly in respect of the DGF incidence. DGF was significantly more frequent in patients with obesity than in overweight (*p* = 0.0015) and normal weight patients (*p* = 0.0005). There was no difference between overweight and normal-weight patients (*p* = 0.86).

In the study group, there were 86 cases of adverse events requiring reoperation within 30 days after transplantation (19.86%).

The short-term surgical complications, reoperation rate, incidence of lymphocele, vascular complications and wound dehiscence were more frequent in obese KTRs. The analysis of postoperative adverse events stratified by type and the comparison between BMI cohorts are reported in Table 2.

The function of the transplanted kidney one month after KT was inferior in patients with obesity. A comparison of the concentration of serum creatinine one month after KT showed significantly higher levels between all groups (*p* < 0.05), between normal versus overweight (*p* = 0.014), normal versus obese (*p* = 0.027) and overweight versus obese (*p* = 0.000), as shown in Table 1. Superior function of the transplanted kidney was observed in normal weight subjects (Table 1).

### 3.1. Mortality and Graft Loss after Kidney Transplantation in BMI Stratified Groups of Patients

Mean observation time was 2.15 years (range from 3 to 7 years) after KT. In this observation period, 33 (7.6%) deaths were reported. Based on above results in normal weight and overweight KTRs groups, long term results of kidney transplantation were analyzed in two groups: non-obese (BMI < 30 kg/m^2^) and obese (BMI ≥ 30 kg/m^2^) KTRs. The mortality rate did not differ between obese and non-obese KTRs: (4.3% in obese and 8.3% non-obese respectively, *p* = 0.19). There were 43 (9.9%) graft losses in follow-up. The incidence of graft loss was higher in obese patients: 17.1% versus 8.5 (*p* = 0.029). The most frequent cause of early graft loss during first year of observation was artery or vein thrombosis that accounted for 55% of recorded graft losses. The number of graft losses was higher in obese versus overweight KTRs: 17.1% in obese and 7.9% in overweight (*p* = 0.009).

In survival analysis, patients with obesity did not differ significantly with respect to 1-year patient survival compared to non-obese (normal weight and overweight) (98.5 vs. 97.8%, respectively, *p* = 0.56), and 1-year graft survival (non-censored for death)—87.1% vs. 92.6%, *p* = 0.11. One-year death-censored graft survival was better in non-obese patients (88.6 vs. 94.8% *p* = 0.05).

### 3.2. Univariate and Multivariate Analysis of Patient Death and Graft Loss

Based on univariate analysis, factors significantly associated with death-censored graft loss were acute rejection episode (ARE), DGF and BMI > 30. Among these factors, only ARE was a significant predictor of graft loss on multivariate analysis (Table 3).

Based on univariate analysis, the age and CCI were associated with mortality. CCI was also an independent predictor of death based on multivariate analysis (OR 1.521 (1.107–2.09) (*p* = 0.01). BMI ≥ 30 was not a predictor of patient mortality (OR 0.505 (0.15–1.702), *p* = 0.27) (Table 4).

## 4. Discussion

Our study provides an exceptionally broad insight into the effect of recipient’s BMI on both intraoperative and perioperative parameters in a large population of KTRs. This study is unique because, until now, little has been known about intraoperative and perioperative outcomes of KT in different BMI groups [7,9]. Our study demonstrates that the percentage of KTRs with obesity reached 16%, whereas patients with overweight constituted 36.8% of KTRs. These results reflect the trend of increasing incidence of overweight and obese KTRs observed in the literature [10,11].

Longer total procedure time and warm ischemia time are observed in patients with obesity. The results of the present study demonstrate that higher BMI (≥30 kg/m^2^) was associated with significantly longer total KT procedure time. Excessive BMI make KT procedure longer, technically more demanding and represent a challenge for the transplant surgeon than in normal BMI patients. What is more, this association was especially observed in males, which may be related to higher incidence of abdominal type of obesity than in females. Interestingly, we did not observe a statistically significant difference in total procedure time between normal BMI and overweight KTRs (Figure 1). Overweight patients seemed to have total procedure time comparable to normal BMI patients. This may indicate the limit of pretransplant weight loss graded by BMI to <30 kg/m^2^ which can be found in current guidelines [8,12]. Significantly longer total procedure time was observed only in the obesity group, which may support the concept of pretransplant weight loss discussed in the literature [10]. Pre-transplant qualification to metabolic surgery in treatment of obesity increases access to kidney transplantation [13].

Secondly, we showed that BMI ≥ 30 is associated with longer warm ischemia time during the transplantation procedure. Obese KTRs have significantly longer warm ischemia times than normal and overweight patients. It was previously shown that long warm ischemic injury during vascular anastomoses in obese KTRs may contribute to induction of chronic allograft nephropathy, interstitial fibrosis, tubular atrophy and graft loss [14,15]. In the context of these data, prolonged warm ischemia present in obese KTRs is related to higher risk of chronic allograft disfunction.

In our analysis the function of the transplanted kidney was significantly inferior in patients with obesity (creatinine concentrations were 1.76 and 1.53 mg/dL in obese and non-obese respectively, *p* = 0.0001) after one-month observation. The best kidney function one month after KT was observed in normal weight recipients. The effect of prolonged warm ischemia time may be exacerbated by calcineurin inhibitors used in post-transplant immunosuppression [16]. Due to this fact, adequate immunosuppressive therapy may be necessary in prevention of chronic nephropathy in obese KT recipients.

There was no difference in the incidence of ARE between the BMI cohorts. The immunological risk of the patient was not related to pretransplant BMI. The most important factors related to calculation of immunological risk were the number of human leukocyte antigen mismatches, sensitization based on panel reactive antibodies, recipient age and immunosuppression regiments. In our analysis, we found a higher incidence of postoperative DGF in excessive BMI patients: 52.1% in obese and 30.1% in overweight KTRs. In BMI group analysis, a significant difference was found between the obese and normal (*p* = 0.0005) and obese and overweight KTRs (*p* = 0.0015) groups, whereas DGF rate was similar in normal and overweight KTRs (*p* = 0.86) groups. Large metanalyses confirm that BMI was found to be an independent predictor of a higher rate of DGF [6,17]. Another recent metanalysis (of 209,000 patients) of KTRs with BMI < 30 kg/m^2^ revealed lower mortality, incidence of DGF, AR, infectious complication rate and better 1-, 2- and 3-year graft survival [7]. Additionally, DGF was found to be an independent risk factor of 1-year graft loss in metanalysis [18]. This supports the necessity of pretransplant obesity treatment in ESKD patients to lower the risk of DGF and improve graft survival.

Despite no difference in total hospitalization time between BMI cohorts in our study, in the literature obese KTRs produce higher costs of transplantation because of higher direct costs of procedure and higher readmission rates [19].

Adverse events were proved to be more frequent in obese patients. Our study demonstrates that short term surgical complications, vascular complications, impaired wound healing, incidence of lymphocele and reoperation rate were more frequent in obese than in overweight and normal weight recipients (Table 2). Additionally, the rate of surgical complications may be impacted by immunosuppression protocol. In the literature, cyclosporin A is related to higher risk of post-transplant bleeding than tacrolimus, and mTOR inhibitors are related to a higher incidence of lymphocele [20,21]. The bariatric procedure before KT is associated with significant changes of pharmacokinetics of tacrolimus and mycophenolate mofetil (increased maximal concentration and decreased clearance) [13]. Both obesity and weight loss related to metabolic surgery should be considered in planning the immunosuppression protocol in patients with obesity.

Early surgical complications after KT are usually emergent and need a rapid response to prevent the early graft loss [22,23]. They are commonly associated with reoperation and can severely affect overall graft survival [22]. Most of them are related to vascular complications, which in our cohort were observed in 17.5% of KTRs. In our analysis, early surgical complications and specific vascular complications were more common in obese KTRs (Table 2). These may expose patients with obesity to higher risk of early graft loss. In the literature, obesity (*p* = 0.0007) and the occurrence of intraoperative complications (*p* = 0.026) were independent risk factors of early graftectomy because of the graft failure [23].

Lymphocele is associated with pain and reduced renal function. Higher incidences in obese than overweight and normal weight may necessitate ultrasound guided aspiration and reoperation to drain the large fluid collection to prevent the deterioration of KT function and infection [24,25].

There is also significant association between obesity and inferior graft survival in KTRs (*p* = 0.029), but only in univariate analysis. In multivariate analysis, there was no significant difference (Table 3). These results are comparable with those in the literature, which makes our cohort representative. In our analysis BMI was related to inferior allograft function at discharge. These may be the result of higher post-transplant DGF incidence and higher rate of surgical complications that are related to decline in kidney function. We believe that in depth analysis of the quality of donors, as well as detailed analysis of comorbidities of the recipients, would enable improved understanding of the actual effect of BMI on the early and late results of KT.

The higher incidence of postoperative adverse events and inferior renal function in patients with obesity supports the need for pretransplant weight loss to improve the results of transplantation and reduce treatment costs. Our data highlight the course of transplantation of overweight is comparable to normal BMI KT recipients, and we believe that pretransplant weight reduction to the BMI < 30 kg/m^2^ might allow a better short-term course of transplantation as well as long-term graft survival. Previous studies support the advantage of weight reduction in this group of patients [26,27]. In our previous study we proved that metabolic surgery was safe and efficient in ESKD patients [28]. Moreover, our long-term observation of graft and patient survival confirmed inferior results of transplantation in obese patients. Greater effort should be made to improve the results of KT in KTRs with obesity to improve the potential benefits of KT in that group of patients and minimize the risk.

There are considerable limitations of our study. The study was single-center, which may limit the generalizability of the results. The nature of the study was a retrospective analysis of a prospectively build database. To minimize selection bias, we included data from all consecutive transplanted patients from our institutional registry. Due to national guidelines, patients with BMI > 35 kg/m^2^ should not be qualified for KT at all. Thus, our cohort of obese KTRs may be not representative if used in inter-study comparisons. Third, we used patients’ BMI, which may be inappropriate in assessing obesity in ESKD patients. Due to preoperative fluid overload, and type of pretransplant renal replacement therapy, BMI may not be reflective in obesity grading in that group of patients. What is more, the number of individual complication types might have been too low to allow for sufficiently powered analysis. Thus, conclusions related to this matter cannot be drawn from this study.

## 5. Conclusions

In summary, kidney recipient’s BMI affects significantly the intraoperative and postoperative course of the KT procedure. Patients with obesity have longer total procedure time and warm ischemia compared to normal and overweight KTRs. The function of the transplanted kidney is inferior in obese KTRs, they are at higher risk of DGF, and postoperative adverse events commonly associated with reoperation. The procedure of kidney transplantation and early postoperative course is more demanding in KTRs with obesity. Graft loss is more frequent in patients with obesity. These data suggest benefits of pretransplant weight loss in ESKD patients, aimed at achieving a body weight within at least the overweight limit to improve both early and long-term results of KT.

## Figures and Tables

**Figure 1 jcm-11-04393-f001:**
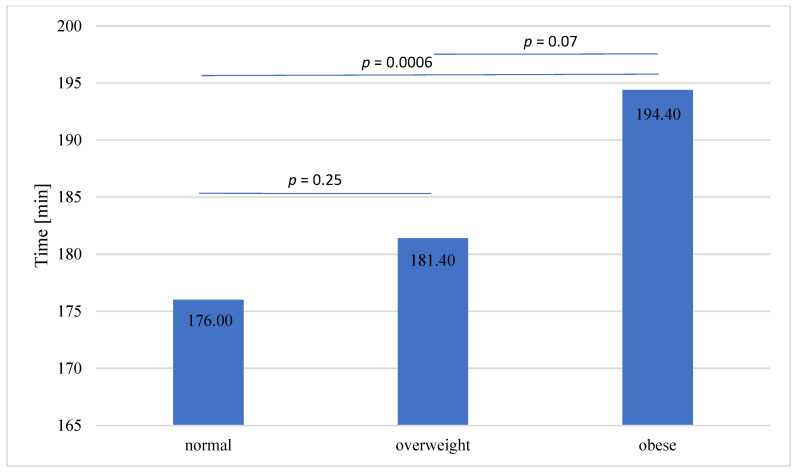
Total procedure time in BMI stratified groups.

**Figure 2 jcm-11-04393-f002:**
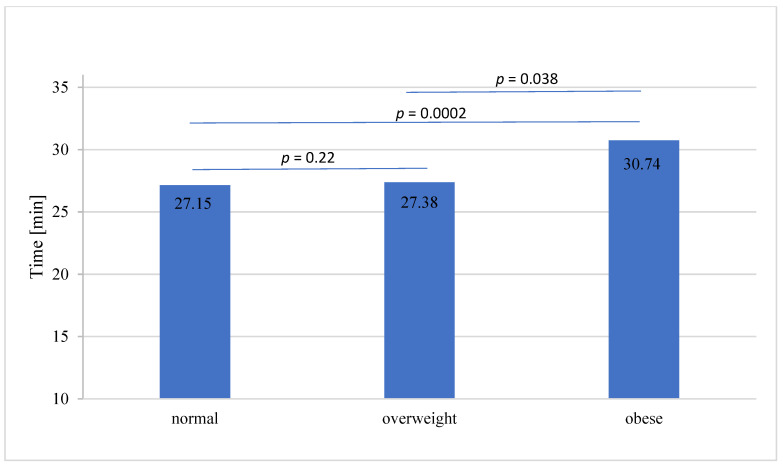
Warm ischemia time in BMI stratified groups.

**Table 1 jcm-11-04393-t001:** Baseline characteristics of all studied groups of patients, the kidney transplantation procedure, and the comparison between normal (BMI < 25), overweight (BMI 25–30) and obese (BMI ≥ 30) kidney recipients.

Variables	All Patients (*n* = 433)	Normal BMI 18.5–24.9 (*n* = 208)	Overweight BMI 25–29.9 (*n* = 153)	Obese BMI ≥ 30 (*n* = 72)	*p*-Value
**Donor**					
**Age**	48.6 (51)	47.1 (49)	49.8 (51)	50.9 (52)	*p* = 0.05
**Body weight (kg)**	77.9 (76)	76.7 (75)	78.1 (75)	80.7 (80)	*p* = 0.06
**Body height (cm)**	173.1 (174)	173.1 (174)	172.9 (173.5)	173.5 (174)	*p* = 0.95
**BMI [kg/m^2^]**	25.9 (24.9)	25.5 (24.7)	26.1 (24.9)	26.7 (26)	*p* = 0.08
**Donor/recipient BMI ratio**	1 (1)	1.2 (1.2)	0.9 (0.9)	0.8 (0.8)	*p* < 0.05
**Donor/recipient body weight ratio**	1.1 (1.1)	1.3 (1.2)	0.9 (1)	0.9 (0.8)	*p* < 0.05
**Recipient**					
**Male (%)**	272 (62.8)	115 (55.3)	114 (74.5)	42 (58.3)	normal vs. overweight *p* = 0.0002 normal vs. obese *p* = 0.65 overweight vs. obese *p* = 0.014
**Female (%)**	161 (37.2)	93 (44.7)	39 (25.5)	30 (41.7)	
**BMI (** **kg/m^2^)**	25.3	21.97	27.22	31.71	*p* < 0.05
**Mean age (median; years)**	49.3 ± 13.8 (51 ± 11.5)	45.3 ± 14.4 (46 ± 12)	52.7 ± 11.6 (55 ± 9)	53.3 ± 13.1 (57 ± 8.5)	normal vs. overweight *p* = 0.00 normal vs. obese *p* = 0.00 overweight vs. obese *p* = 1
**Mean Charlson Comorbidity Index (median)**	3.4 ± 1.4 (3 ± 1)	3.14 ± 1.2 (3 ± 1)	3.6 ± 1.4 (4 ± 1.5)	3.86 ± 1.7 (3 ± 1.5)	*p* = 0.029 normal vs. overweight *p* = 0.01 normal vs. obese *p* = 0.01 overweight vs. obese *p* = 1
**Comorbidities**					
**Hypertension**	322 (74.3)	147 (70.3)	121 (76.1)	54 (83.1)	*p* = 0.09
**Diabetes Mellitus type 1 and**	64 (14.8)	16 (7.7)	31 (19.5)	17 (26.2)	*p* = 0.0001
**Coronary artery disease**	65 (15)	30 (13.4)	21 (13.2)	14 (21.5)	*p* = 0.26
**Other heart diseases ^1^**	44 (10.2)	26 (12.4)	13 (8.2)	5 (7.7)	*p* = 0.31
**Benign prostate hyperplasia**	22 (5.1)	7 (3.3)	11 (6.9)	4 (6.2)	*p* = 0.27
**Thyroid disease**	32 (7.4)	15 (7.2)	12 (7.5)	5 (7.7)	*p* = 0.98
**Parathyroid disease**	47 (10.9)	18 (8.6)	24 (15.1)	5 (7.7)	*p* = 0.09
**Pulmonary disease ^2^**	30 (6.9)	14 (6.7)	10 (6.3)	6 (9.2)	*p* = 0.72
**Cerebral stroke**	17 (3.9)	8 (3.8)	7 (4.4)	2 (3.1)	*p* = 0.89
**Digestive track diseases ^3^**	72 (16.6)	28 (13.4)	28 (17.6)	16 (24.6)	*p* = 0.1
**Neoplasms history**	29 (6.7)	13 (6.2)	11 (6.9)	5 (7.7)	*p* = 0.91
**Hepatitis infection** **Active tobacco abuse**	34 (7.9) 10 (2.3)	21 (10) 4 (1.9)	12 (7.5) 2 (1.3)	1 (1.5) 4 (6.2)	*p* = 0.08 *p* = 0.08
**ESKD etiology**					n/a
**Glomerulonephritis**	151 (35.6)	83 (39.9)	49 (32)	19 (26.4)	
**Diabetic nephropathy**	52 (12.3)	18 (8.7)	22 (14.4)	12 (16.7)	
**Hypertensive nephropathy**	38 (8.9)	11 (5.3)	20 (13.1)	7 (9.7)	
**Interstitial nephropathy**	49 (11.6)	28 (13.5)	16 (10.5)	5 (6.9)	
**ADPKD**	56 (13.2)	19 (9.1)	28 (18.3)	9 (12.5)	
**Other ^4^**	22 (5.2)	10 (4.8)	6 (3.9)	6 (8.3)	
**Unknown**	56 (13.2)	25 (12)	23 (15)	8 (11.1)	
**Dialysis modality before KT (%)**	HD 329 (76) PD 70 (16.2) PREE 34 (7.8)	HD 151 (72.6) PD 34 (16.3) PREE 14 (6.7)	HD 120 (78.4) PD 27 (17.6) PREE 16 (10.4)	HD 58 (82.8) PD 9 (11.4) PREE 4 (5.7)	*p* = 0.09 *p* = 0.16 *p* = 0.33
**Transplantation**					
**2nd and 3rd KT (%)**	59 (13.6)	38 (18.2)	16 (10.5)	5 (6.9)	*p* = 0.056 normal vs. overweight *p* = 0.04 normal vs. obese *p* = 0.02 overweight vs. obese *p* = 0.52
**Total procedure time, mean (median; min)**	181.98 (180)	176 ± 36.2 (180 ± 22.5)	181.4 ± 36.1 (180 ± 17.5)	194.4 ± 38.9 (195 ± 30.0)	*p* = 0.0025
**WIT mean (median; min)**	27.83 ± 9.3 (26 ± 4.5)	27.15 ± 10.6 (25 ± 4.5)	27.38 ± 7.0 (27 ± 4.5)	30.74 ± 9.2 (30.0 ± 5.5)	*p* = 0.0003
**CIT mean (median; min)**	922.6 ± 636 (894 ± 223)	927.0 ± 370 (903 ± 206)	899.5 ± 343 (863 ± 215)	958.7 ± 389 (935 ± 236)	*p* = 0.1509
**Post-transplant hospitalization**					
**Serum creatinine mean (median) one month after KT (mg/dL)**	1.575 ± 0.571 (1.52 ± 0.34)	1.474 ± 0.527 (1.360 ± 0.345)	1.596 ± 0.502 (1.545 ± 0.335)	1.838 ± 0.742 (1.750 ± 0.225)	*p* = 0.000 normal vs. overweight *p* = 0.014 normal vs. obese *p* = 0.027 overweight vs. obese *p* = 0.000
**AR (%)**	42 (9.7)	21 (10.04)	13 (8.5)	8 (11.3)	*p* = 0.36
**DGF (%)**	144 (33.2)	61 (29.2)	46 (30.1)	37(52.8)	*p* = 0.0002
**Total hospitalization time (days)**	21.57 ± 12.3 (18 ± 5.5)	20.19 ± 11.3 (18 ± 4.5)	21.87 ± 12.6 (19 ± 6.0)	25.04 ± 13.6 (20 ± 8.5)	*p* = 0.1509

Abbreviations: HD—hemodialysis. PD—peritoneal dialysis. PREE—preemptive transplantation and retransplantation. KT—kidney transplantation. AR—acute rejection. DGF—delayed graft function. WIT—warm ischemia time. CIT—Cold ischemia time. ^1^ Other heart diseases including: arrhythmias, valvular defects. ^2^ chronic obstructive pulmonary disease, asthma, sarcoidosis. ^3^ Peptic ulcer disease, gastroesophageal reflux disease, diverticulosis, inflammatory bowel disease. ^4^ Other causes of ESKD including: hemolytic uremic syndrome. amyloidosis. ESKD after nephrectomy because of cancer or after chemotherapy).

**Table 2 jcm-11-04393-t002:** Postoperative adverse events of kidney transplantation stratified by type and BMI (*p* < 0.05 statistically significant).

Adverse Event	All Patients (*n* = 433)	Normal (*n* = 209)	Overweight (*n* = 153)	Obese (*n* = 71)	*p*-Value
**Postoperative kidney function**					
Acute rejection (%)	42 (9.7)	21 (10.04)	13 (8.5)	8 (11.3)	*p* > 0.05
Delayed graft function (%)	144 (33.2)	61 (29.2)	46 (30.1)	37 (52.1)	obese vs. normal *p* = 0.0005 obese vs. overweight *p* = 0.0015 normal vs. overweight *p* = 0.86
**General adverse events**					
Cardiological complications ^1^ (%)	20 (4.6)	9 (4.3)	8 (5.2)	3 (4.2)	*p* > 0.05
Infectious complications ^2^ (%)	106 (24.5)	44 (21.0)	41 (26.8)	21 (29.6)	*p* > 0.05
Posttransplant diabetes (%)	46 (10.6)	19 (9.1)	22 (14.4)	5 (7.0)	*p* > 0.05
**Surgical adverse events**					
Surgical complications < 30 days (%)	111 (25.65)	53 (25.3)	27 (16.3)	31 (43.7)	obese vs. normal *p* = 0.0036 obese vs. overweight *p* = 0.00001 normal vs. overweight *p* = 0.08
Surgical complications > 30 days (%)	24 (5.5)	10 (4.8)	7 (4.6)	7 (9.8)	*p* > 0.05
Reoperation (%)	86 (19.9)	37 (17.7)	23 (15.0)	26 (36.6)	obese vs. normal *p* = 0.001 obese vs. overweight *p* = 0.0003 normal vs. overweight *p* = 0.4997
Lymphocele (%)	51 (11.8)	17 (8.1)	15 (9.8)	19 (26.8)	obese vs. normal *p* = 0.0001 obese vs. overweight *p* = 0.01 normal vs. overweight *p* = 0.58
Urological complications (%)	32 (7.4)	18 (8.6)	7 (4.6)	7 (9.8)	*p* > 0.05
Wound dehiscence (%)	21 (4.8)	2 (0.96)	7 (4.6)	12 (16.9)	obese vs. normal *p* = 0.0000 obese vs. overweight *p* = 0.0021 normal vs. overweight *p* = 0.0329
Vascular complications (%)	76 (17.5)	42 (20.1)	15 (9.8)	19 (26.8)	obese vs. normal *p* = 0.2398 obese vs. overweight *p* = 0.001 normal vs. overweight *p* = 0.0079

^1^ Myocardial infarct. atrial fibrillation. heart failure. ^2^ Urinary tract infections, pneumonia, CMV infection.

**Table 3 jcm-11-04393-t003:** Univariate and Multivariate Analysis of predictors of Graft’s Loss (Death Censored) in patients after kidney transplantation.

Variable	Univariate Analysis, OR (95% CI)	*p*-Value	Multivariate Analysis, OR (95% CI)	*p*-Value
Age (y)	0.997 (0975–1.02)	0.8	-	-
Gender (F/M)	2.2 (1.073–4.55)	0.69	-	-
Charlson comorbidity index	0.84 (0.664–1.076)	0.16	-	-
HD before KT	1.221 (0.565–2.637)	0.6	-	-
BMI	1.05 (0.973–1.131)	0.21	-	-
BMI > 30	2.209 (1.073–4.55)	0.04	1.9 (0.898–4.035)	0.09
ARE	2.293 (1.215–4.327)	0.005	2.9 (1.295–6.649)	0.01
DGF	2.293 (1.215–4.327)	0.001	1.84 (0.947–3.576)	0.07
WIT	1.019 (0.978–1.061)	0.4	-	-
CIT	1.0 (0.999–1.001)	0.76	-	-
KT number 1	1.051 (0.423–2.613)	0.42	-	-

Abbreviations: ARE—acute rejection episode; BMI—body mass index; CIT—cold ischemia time; DGF—delayed graft function; HD—hemodialysis; KT—kidney transplantation; WIT—warm ischemia time.

**Table 4 jcm-11-04393-t004:** Univariate and Multivariate Analysis of predictors of mortality after kidney transplantation.

Variable	Univariate Analysis, OR (95% CI)	*p*-Value	Multivariate Analysis, OR (95% CI)	*p*-Value
Age (y)	1.079 (1.04–1.119)	0.00	1.04 (0.994–1.087)	0.088
Gender (F/M)	1.167 (0.055–2.474)	0.686	-	-
Charlson comorbidity index	1.833 (1.442–2.33)	0.00	1.521 (1.107–2.09)	0.01
HD before KT	1.465 (0.588–3.653)	0.412	-	-
BMI	0.983 (0.902-1.07)	0.691	-	-
BMI > 30	0.505 (0.15-1.702)	0.27	-	-
PD before KT	0.918 (0.342–2.464)	0.863	-	-
KT number 2 or 3	0.863 (0.292–2.549)	0.786	-	-
Serum creatinine concentration mg/dL one month after KT	1.389 (0.741–2.601)	0.305	-	-
Kidney graft loss	0.921 (0.269–3.156)	0.894	-	-

Abbreviations: BMI—body mas index, KT—kidney transplantation, HD—hemodialysis, PD—peritoneal dialysis.

## Data Availability

Not applicable.

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
