# Peer review of "Pretransplant BMI Significantly Affects Perioperative Course and Graft Survival after Kidney Transplantation: A Retrospective Analysis"

_jcm, 2022, doi:10.3390/jcm11154393_

Round 1
Reviewer 1 Report
This is a nice single center analysis of kidney transplant outcomes related to recipient BMI; obese patients had worse outcomes. Unfortunately, this is not a novel observation, and confirms previously published registry analyses with thousands of patients.
Author Response
Dear Reviewer,
We thank the Reviewer for revision of the manuscript. We agree that the impact of recipient BMI on posttransplant kidney function has been studied previously but the main focus of current study was to analyse intraoperative and postoperative course of kidney transplantation procedure and postoperative complications rate in three cohorts based on kidney transplant recipients BMI. The novelty of our study is the fact that the relation of BMI to analysed parameters: the course of transplantation procedure (total procedure time, warm ischemia time), and perioperative course (complications of the procedure, perioperative adverse events, kidney function one month after the procedure, hospitalization time) were not previously, to our knowledge, studied and published in literature.
We corrected linguistic and grammatical errors as suggested by the reviewer.
We hope that the novelty of the intraoperative results and the interesting association of BMI to intraoperative and postoperative parameters which were not previously published in the literature will be accepted for publication in Journal of Clinical Medicine.
On behalf of the Authors,
Małgorzata Dobrzycka
Mail [email protected]
Phone +48 603647525
Department of General, Endocrine and Transplant Surgery,
Medical University of Gdańsk,
Skłodowskiej 3A Str
80-210 Gdańsk, Poland
Reviewer 2 Report
I read this paper and could not detect any major mistakes. Besides the graphs which should look more professional I agree with the current form.
The tables should be more professional.
I added a kidney transplant paper as an example to may improve the current form, also some citations especially with the associated complications seems not be the latest.
https://pubmed.ncbi.nlm.nih.gov/34640413/
Author Response
Dear Reviewer,
We thank the Reviewer for revision of the manuscript and important comments.
The tables and graphs were improved according to your recommendation. In discussion section we include the latest citations on complications of kidney transplantation procedure. Those citations were presenting the latest update in surgical complications of kidney transplantation procedure. The novelty of our study is the fact that the relation of BMI to the course of transplantation procedure (total procedure time, warm ischemia time), and perioperative course (complications of the procedure, perioperative adverse events, kidney function one month after the procedure, hospitalization time) were not previously, to our knowledge, studied and published in literature.
We corrected linguistic and grammatical errors as suggested by the reviewer.
We hope that the novelty of the intraoperative results and the interesting association of BMI to intraoperative and postoperative parameters which were not previously published in the literature will be accepted for publication in Journal of Clinical Medicine.
On behalf of the Authors,
Małgorzata Dobrzycka
Mail [email protected]
Phone +48 603647525
Department of General, Endocrine and Transplant Surgery,
Medical University of Gdańsk,
Skłodowskiej 3A Str
80-210 Gdańsk, Poland
Reviewer 3 Report
The increasing obesity epidemic around the world and especially in both transplant kidney donor and recipients makes this topic very relevant. I will recommend that the authors help clarify the following for proper understanding of the research work:
1.Pre transplant immunologic risk in all BMI group to help understand if contributory to higher risk of Acute rejection in those with BMI > 40
2. Any difference in the perioperative outcome depending on the severity of obesity?
3. Comment on the generalization of this findings in other parts of the world with BMI > than the average of 25 reported in the study.
4. Despite the identified perioperative complications associated with obesity, author should comment on the potential benefit of kidney transplant in this population compared to remaining on dialysis.
5. While recommending the pre transplant weight loss in the appropriate patient, the author should comment if that should be an absolute requirement for kidney transplant wait-listing.
6. Comment on if the identified perioperative findings affects long term survival (3 years+++) as one one year survival data was reported.
Author Response
Dear Reviewer,
We thank the Reviewer for important comments. The main focus of the study was to analyse intraoperative and postoperative course of kidney transplantation procedure and postoperative complications rate in three cohorts based on kidney transplant recipients BMI. The novelty of our study is the fact that the relation of BMI to analysed parameters: the course of transplantation procedure (total procedure time, warm ischemia time), and perioperative course (complications of the procedure, perioperative adverse events, kidney function one month after the procedure, hospitalization time) were not previously, to our knowledge, studied and published in literature.
In response to your comments the immunological risk in obese kidney recipients was not higher than the normal weight based on literature review. In our material there is no difference in the incidence of ARE between the BMI cohorts also. The most important factors related to calculation of immunological risk were the number of human leukocyte antigen mismatches, sensitization based on panel reactive antibodies, recipient age and immunosuppression regiments. We include additional statement in Discussion section.
The difference in the perioperative outcome depending on the severity of obesity can not be the focus of our study because of the fact that kidney transplant recipients with BMI higher than 35 kg/m2 were excluded from the transplantation according to our national and institutional guidelines.
We agree that several efforts should be made to improve the results of KT in KTRs with obesity to improve the potential benefits of KT in that group of patients. Often due to obesity associated comorbidities their access to the transplant list and transplantation is inferior than non-obese. Bariatric surgery is considered as a safe method of obesity treatment in that specific group of patients, and feasible as a bridge therapy to kidney transplantation. Additionally, weight loss after bariatric surgery may improve kidney function to an acceptable level, and delay the qualification to dialysis therapy in patients undergoing bariatric surgery before ESKD development. It was proven that weight loss resulted in an improvement of proteinuria, albuminuria and normalization of the glomerular filtration rate. For dialyzed ESKD-patients bariatric surgery can slow the progression and allow them to stay on the transplant waiting list. It is also worth to mention about the “reverse epidemiology” of obesity in ESKD patients and a lower risk of death in obese ESKD patients.
Based on our study surgical results we did not observe a statistically significant difference in total procedure time and warm ischemia time between normal BMI and overweight KTR as shown in Table 1 and Figures 1 and 2. Those findings may indicate the limit of pretransplant weight loss graded by BMI to <30 kg/m2 but we cannot recommend it as a guideline now. Other studies are needed to clearly indicate the pretransplant weight loss limit and include it in kidney transplant guidelines.
The analysis of survival was not the main focus of our study. Additionally to surgical results to make our study more representational to comparison to other studies we preformed the survival analysis. In our cohorts patients with obesity did not differ significantly with respect to 1-year patient survival compared to non-obese (normal weight and overweight) (98.5 vs 97.8%, respectively, p=0.56), and 1-year graft survival (non-censored for death) – 87.1% vs 92.6%, p=0.11. 1-year death-censored graft survival was better in non-obese patients (88.6 vs 94.8% p=0.05). Those results were comparable with literature. In long term survival analysis the literature presents the comparable survival of obese and non-obese KT recipients despite the higher risk of surgical complications and DGF.
The Reviewer questions were addressed in the Discussion section. It would be interesting to perform the in depth analysis of all of them to improve the results of kidney transplantation.
We hope that the novelty of the intraoperative results and the interesting association of BMI to intraoperative and postoperative parameters which were not previously published in the literature will be accepted in the revised version of our manuscript and will be suitable for publication in Journal of Clinical Medicine.
On behalf of the Authors,
Małgorzata Dobrzycka
Mail [email protected]
Phone +48 603647525
Department of General, Endocrine and Transplant Surgery,
Medical University of Gdańsk,
Skłodowskiej 3A Str
80-210 Gdańsk, Poland
Round 2
Reviewer 1 Report
The paper is better, in terms of the English, but again, the novelty is low. It is odd in their table that the mean BMI of the "obese" patients is 26, which is not obese.
The authors talk about the need for patients to have a lower BMI prior to getting transplanted. The reality is that few patients want to undergo bariatric surgery, and that the mortality advantage of being transplanted is vastly better than the slight decrement in outcomes.
Author Response
Dear Reviewer,
We thank the Reviewer for comments. We agree with the reviewer, that there is no doubt, that long term results of Ktx were well documented in obese patients’ cohorts, however board evaluation including technical aspects of KTx with long term results was never previously published. The main focus of the study was to analyse intraoperative and postoperative course of kidney transplantation procedure and postoperative complications rate in three cohorts based on kidney transplant recipients BMI. Main authors are surgeons, the novelty of the study is the presentation of the perioperative outcomes including the course of transplantation procedure (total procedure time, warm ischemia time), perioperative course (complications of the procedure, perioperative adverse events, kidney function one month after the procedure, hospitalization time). It was first time shown that patients with obesity were technically more demanding during transplantation procedure (longer warm ischemia time and longer total transplantation procedure, the two main results of the study presented in Table 1 and Figures 1 and 2) but also that those results differs only between obese and normal weight cohost with no difference between normal weight and overweight. This observation in the era of obesity in ESKD patients may have an impact on the limit of preoperative weight loss in obese KTx recipients nor qualified to the transplantation because of the obesity.
In our study we raise a subject of pretransplant weight loss in obese ESKD patients. According to our national guidelines patients with BMI>35 kg/m2 were not qualified to kidney transplantation at all until they lose weight. In our center weight loss surgeries are being currently offered to all patients who do not qualify to Tx because of BMI above national guidelines limit. Therefore the weight loss surgeries are safe and are offering effective weight loss not only improves outcomes but for some patients makes Tx formally possible. We have severally published that this approach is both feasible and safe in that group of patients1–4.
- Dobrzycka M, Proczko-Stepaniak M, Kaska Ł, Wilczyński M, Dębska-Ślizień A, Kobiela J. Weight Loss After Bariatric Surgery in Morbidly Obese End-Stage Kidney Disease Patients as Preparation for Kidney Transplantation. Matched Pair Analysis in a High-Volume Bariatric and Transplant Center. Obes Surg. 2020. doi:10.1007/s11695-020-04555-8
- Proczko M, Kaska Ł, Kobiela J, Stefaniak T, Zadrożny D, Śledziński Z. Bariatric surgery in morbidly obese patients with chronic renal failure, prepared for kidney transplantation--case reports. Pol Przegl Chir. 2013;85(7):407-411. doi:10.2478/pjs-2013-0062
- Proczko M, Kaska Ł, Kobiela J, Stefaniak T, Zadrozny D, Śledziński Z. Roux-en-Y gastric bypass in dialysed morbidly obese patients as a preparation for a kidney transplantation: Case series. Wideochirurgia I Inne Tech Maloinwazyjne. 2013;8(2):174-177. doi:10.5114/wiitm.2011.32852
- Małgorzewicz S, Dębska-Slizień A, Czajka B, Owczarzak A, Rutkowski B. Influence of Body Mass on Kidney Graft Function in Patients After Kidney Transplantation. Transplant Proc. 2016;48(5):1472-1476. doi:10.1016/J.TRANSPROCEED.2015.12.137
In first paragraph of the Table 1 we present the information about the donors. That is why in obese kidney recipients column the mean BMI is 26.7 kg/m2 which is the donor BMI in the obese recipient group. The obese recipient cohort mean BMI is 31.71 kg/m2. We present the significant difference in donor recipient BMI ratio (p=0.00). Those result were presented in Table 1 and the Results section of the manuscript.
We hope that the novelty of the intraoperative results and the interesting association of BMI to intraoperative and postoperative parameters which were not previously published in the literature will be accepted by the reviewer and will be suitable for publication in Journal of Clinical Medicine.
On behalf of the Authors,
Małgorzata Dobrzycka
Mail [email protected]
Phone +48 603647525
Department of General, Endocrine and Transplant Surgery,
Medical University of Gdańsk,
Skłodowskiej 3A Str
80-210 Gdańsk, Poland
This manuscript is a resubmission of an earlier submission. The following is a list of the peer review reports and author responses from that submission.
Round 1
Reviewer 1 Report
At the line 42 authors affirmed that “obesity increases the risk for diabetes and hypertension de novo diagnosis after transplantation” that is a correct sentence.
Since in the remaining part of the article there is not additional focus on this topic, the beforementioned sentence seems to be not correlated.
At the line 58-59, in order the sentence to be more comprehensible, it could be explained the technique used for the surgery.
Results are well structured with good graphics. We suggest to insert the charts before their explication. From line 160 to 163 it is unclear the comparison between overweight and normal weight.
In the discussion, some changes are necessary: at line 203-204 “the results of the present study demonstrates the higher BMI (>=25kg/m2)…” is in contrast with what declared along all the article (total KT procedure time increased for BMI >=30 kg/m2 only).
Studying the function of transplanted kidney, it represents a limit to evaluate the creatinine concentration only. It should be useful to additionally study the creatinine clearance, proteinuria per day, purification capacities.
Moreover, when talking about DGF, it would be interesting to show more information regarding the donor.
To integrate the obesity concept, the circumference of the waist could be taken in to account, as suggested by the KDIGO guidelines.
Reviewer 2 Report
Summary of paper
The purpose of this single-center, observational, retrospective study is to assess whether body mass index at the time of transplant is associated with perioperative course, 1-year graft loss and mortality in 433 cadaveric kidney transplant recipients. The authors provide data suggesting that obesity is related to inferior graft survival. This is an interesting topic, but the current manuscript has significant issues, mainly related to absence of important information. More specifically additional details regarding characteristics of donors at the time of transplant, of donor/recipient body weight and body mass index ratio and on immunosuppressive protocols should be given.
Title and Abstract
Please clarify in the Title that the study was retrospective.
The Authors emphasized in the Abstract the statistically significant association (p=0.04) between body mass index and graft loss at the univariate analysis. However in the multivariable model the p value didn’t reach statistical significance (p=0.09, Table 3). The latter result should be also reported and discussed.
Methods
The Authors assessed that they evaluated the association between body mass index and postoperative adverse events using the Analysis of Variance or Kruskal-Wallis test. This approach is unusual because postoperative adverse events represent the dependent variable and not the grouping factor, while body mass index is the explanatory variable.
The Authors mentioned an ‘F Coxa test’: please clarify this point.
Results
The Authors selected 433 patients enrolled from 2014 to 2017. It is unclear whether they included all available cadaveric kidney transplant recipients or whether they excluded some patients. A flow chart of the study could be added to clarify patients’ selection.
Table 1 should include relevant information on donors (i.e., age, gender, body weight and body mass index) and on donor/recipient body weight and body mass index ratio.
In Table 1 comorbidities are summarized in a single score (Charlson Comorbidity index). However, it would be important to add all single relevant comorbid conditions.
Similarly, Table 1 should include primary cause of renal failure, according to BMI categories.
A table reporting blood pressure and main laboratory parameters at baseline an on follow up should be added.
Information on the immunosuppressive protocols should be given. Of note it would be important to know whether immunosuppression dosing may be sub-optimal in obese patients.
Discussion
The emphasis on the significant association between body mass index and graft loss should be toned down. In the multivariable model the above association was statistically less clear (p=0.09).
Among the limitations the Authors should include that it was a single-center study, which limits the generalizability of the results.
